# A Consolidated Understanding of the Contribution of Redox Dysregulation in the Development of Hearing Impairment

**DOI:** 10.3390/antiox13050598

**Published:** 2024-05-13

**Authors:** Xin Yi Yeo, Soohyun Kwon, Kimberley R. Rinai, Sungsu Lee, Sangyong Jung, Raekil Park

**Affiliations:** 1Department of Psychological Medicine, Yong Loo Lin School of Medicine, National University of Singapore, Singapore 119228, Singapore; xinyi.yeo12@sps.nus.edu.sg; 2Department of Medical Science, College of Medicine, CHA University, Seongnam 13488, Republic of Korea; shkwon2002@gmail.com; 3Department of BioNanotechnology, Gachon University, Seongnam 13120, Republic of Korea; 4Department of Life Science, College of Medicine, CHA University, Seongnam 13488, Republic of Korea; kim.rr@chauniv.ac.kr; 5Department of Otolaryngology-Head and Neck Surgery, Chonnam National University Hospital and Medical School, Gwangju 61469, Republic of Korea; minsunglss@naver.com; 6Department of Biomedical Science and Engineering, Gwangju Institute of Science & Technology (GIST), Gwangju 61005, Republic of Korea

**Keywords:** hearing loss, auditory deficit, cochlea function, redox imbalance, oxidative stress

## Abstract

The etiology of hearing impairment is multifactorial, with contributions from both genetic and environmental factors. Although genetic studies have yielded valuable insights into the development and function of the auditory system, the contribution of gene products and their interaction with alternate environmental factors for the maintenance and development of auditory function requires further elaboration. In this review, we provide an overview of the current knowledge on the role of redox dysregulation as the converging factor between genetic and environmental factor-dependent development of hearing loss, with a focus on understanding the interaction of oxidative stress with the physical components of the peripheral auditory system in auditory disfunction. The potential involvement of molecular factors linked to auditory function in driving redox imbalance is an important promoter of the development of hearing loss over time.

## 1. Introduction

Human perception is an essential biophysical construct of the neurological system, involving the initial detection of environmental stimuli with the sensory organs [1,2,3,4], processing of the spatially dispersed information received and presented as varying local field potentials in the primary sensory cortices [5,6], and the integration of signals within the thalamocortical circuits [7,8], aiding with the identification and interpretation of environmental sensory stimuli and to comprehend the world around an individual. As such, depending on the severity of the primary sensory deficits, individuals experience limitations in physical function [9,10] and neurocognitive changes [11,12,13], which lead to societal costs in the form of economic [14] and psychosocial impacts [15,16]. Of the sensory modalities present in humans, the loss of visual and auditory perception likely has the most significant impact on the overall well-being of an individual in the visual- and audio-dominated modern society [17]. Despite the relatively well-described anatomy of the auditory system [18] and the general process of auditory signal transmission [19,20], there is a lack of understanding of the precise interactions between known environmental, physical, and biomolecular risk factors in driving auditory dysfunction beyond the limited investigation of phenotypic changes based on the previously known molecular functions of the factors.

The redox system is an important component of cellular function and homeostasis. Under physiological conditions, the generation and elimination of reactive molecular species, comprising the radical species—hydroxyl radical (HO^•^), singlet oxygen (^1^O_2_), superoxide (O_2_^•−^), nitric oxide (NO^•^), and peroxynitrite (ONOO^−^)—and non-radical species such as hydrogen peroxide (H_2_O_2_) are actively balanced by the endogenous antioxidant molecules to prevent deleterious effects on the deoxyribonucleotide (DNA) [21], lipid [22], and protein structures [23]. Glutathione (GSH) is the most abundant endogenous antioxidant molecule (1–10 mM range) [24] that can directly react with O_2_^•−^ and alternate antioxidant molecules to reform the antioxidant depleted during radical scavenging [25]. For instance, GSH can reduce the dehydroascorbic acid formed in the conversion of α-tocopheroxyl radical back to the α-tocopherol antioxidant. GSH is synthesized by a two-step process. The first involves the rate-limiting, adenosine triphosphate (ATP)-dependent fusion of L-cystine and L-glutamine to give γ-glutamyl-cisteine catalyzed by the γ-glutamyl-cysteine ligase (GCL), and this is followed by the GSH synthetase-dependent fusion of γ-glutamyl-cisteine with L-glycine to γ-glutamyl-cisteinyl-glicine (GSH).

In the presence of reactive oxygen species (ROS) sources, two molecules of GSH are converted to the oxidized form (GSSG) by glutathione peroxidase (GPx). GSH can be regenerated by the reduction of GSSG with glutathione reductase (GR), flavin adenine dinucleotide (FAD), and nicotinamide-adenine dinucleotide phosphate (NADPH) [26]. The dynamic conversion of GSH to GSSG, involving alternate antioxidants, and vice versa forms the key redox cycle responsible for the regulation of cellular ROS levels. GSH may alternatively react with cysteine residues in proteins, which can be reversed by glutathione S-transferase (GST), or GSSG being exported into the extracellular environment to maintain redox balance. As a further mechanism of redox control, GCL function is inhibited by excess GSH and limited by L-cysteine [27]. Alternatively, thioredoxins and glutaredoxins or alternate proteins containing redox-active disulphide bonds (CXXC) can restore protein function by reversing the oxidation of amino acid residues by ROS, where methionine residue is oxidized to sulfoxide and affects the protein structure [28,29,30]. Excessive, uncontrolled ROS production associated with the dysfunction of the electron transport chain [31], xenobiotic metabolism [32], or inflammatory events [33,34] can deplete the endogenous antioxidants, leading to the development of oxidative stress and damage to the molecular and organelle structures.

The high metabolic demand of sensory hair cells to sustain constant synaptic transmission [35,36] and the dependence of mechanotransduction and auditory function on optimal Ca^2+^ gradients [37,38], maintained through active uptake into the hair cells driven by the mitochondrial potential [39], lead to persistent exposure to reactive oxygen species (ROS), a normal byproduct of oxidative phosphorylation and mitochondrial respiration [40]. It is thought that dysregulation in cellular redox activity results in the net accumulation of ROS, which may lead to molecular and organelle damage [41,42] and, with persistently elevated ROS levels, activation of the apoptotic pathways, cumulating to cellular death [43]. The presence of genetic mutations that alter the direct susceptibility of the auditory system toward the deleterious effects of ROS or the age-dependent accumulation of genetic and molecular damages are suspected drivers of most forms of hearing impairment [44,45,46]. This review takes a slightly different approach from the recent reviews looking at established mechanisms of hearing loss [41,44,47,48] and attempts to build an understanding of the relationship between the known environmental and causal factors of auditory dysfunction and redox dysfunction and how the changes cumulate to hearing impairment. We will also propose potential biological nodes of the auditory process and drugs that may help in delaying or reducing hearing loss pathology.

## 2. Contribution of the Anatomy and Biology of the Auditory System to Its Function

The peripheral auditory system is responsible for the initial detection and encoding of the acoustic input based on the temporal and spectral factors and intensity of the auditory signals received. Hence, overall hearing perception is heavily modulated by the method of sound detection, receiving, processing, and transmission to the central nervous system (CNS) [49] (Figure 1A). The transmission of sound to the cochlea occurs initially via the physical conduction of sound waves through the outer ear structures before reaching the tympanic membrane. The transfer of sound waves occurs through the air column in the ear canal or the solid bone structure and softer tissue structures in the outer ear compartment [50,51,52] (Figure 1B). The sound wave-induced displacement of the tympanic membrane that forms the entrance to the middle ear structure leads to downstream movement of the fenestra ovalis (oval window) through a system of physical pulleys involving the malleus, incus, and stapes (ossicle) bones and the interconnected muscles and ligaments [53,54] (Figure 1C). The sealed bone structure and liquid environment of the cochlea allow the transduction of oval window displacements through the incompressible perilymph within the cochlear duct [55] to the movement of the basilar membrane and part of the organ of Corti (inner ear) [56] (Figure 1D, top). As such, the deflection of the stereocilia on the surface of hair cells physically linked to the moving tectorial membrane opens the mechanosensitive K^+^ channels present on the tip links [57], resulting in the influx of K^+^ from the high-potassium endolymph bath into and consequently depolarizing the hair cell [58]. Alterations in the volume, osmolarity, or ionic composition of the perilymph and endolymph affect the turgor pressure of the hair cells, which alters their capability to produce the electromotive force required for activation of the mechanosensitive channel [59] and synaptic vesicle release machinery, which are key determinants of hair cell and auditory function (Figure 1D). Depolarization of the hair cells activates the voltage-gated calcium channels (Ca_v_), allowing the influx and accumulation of Ca^2+^ within the hair cell and triggering a Ca^2+^-dependent fusion of synaptic vesicles [60,61] for the glutamatergic transmission [62] of auditory signals into the CNS through the auditory nerve.

The outer and inner hair cells differ in their physical locations within the organ of Corti, the types of innervation received from auditory neurons, and their functions. The afferent neurons contact the inner hair cells (IHCs) for the transmission of sound signals into the brain, while the efferent innervations found on the outer hair cells (OHCs) likely contribute to the CNS-dependent modulation of cochlear responses [63]. In addition, a small amount of type II afferent neurons receive inputs from the OHC through en passant synapses [64]. However, these neurons are unlikely to encode significant acoustic information, as they have a low synapse activity level insufficient for neuron activation at basal conditions [65]. Physically, the OHCs mechanically amplify low-level, sound-induced membrane movement through a prestin-dependent mechanism [66], and the IHCs are the primary sensory receptors responsible for auditory signal transmission to the auditory nerve and hence the CNS [20]. The activity of the ribbon synapse is also sensitive toward the frequency, strength, and timing of the sound waves. The presence of varying amounts of large-conductance Ca^2+^-activated K^+^ (BK) channels, which allows the efflux of K^+^ during Ca_v_ activation, likely sharpens frequency detection and tuning by the inner hair cells within the tonotopic map of the cochlea [67], while varying the voltage dependence of the Ca^2+^ influx between individual ribbon synapses likely influences the population of spiral ganglion neurons (SGNs) recruited for signal propagation [68]. The resulting neurochemical signal received by the auditory nerve travels through the acoustic stria [69], superior olivary nucleus [70], and nucleus of lateral lemniscus [71] of the brain stem before passing through the inferior colliculus [72] and medial geniculate nucleus of the midbrain to arrive at the auditory cortex [73,74]. Once in the cortex, the information transmitted is further processed to influence cognition and behavioral outcomes [75,76]. Hence, auditory acuity is affected individually by the direct functional components of the auditory signal transmission process or indirectly through the cross-interactions between different components of the auditory system that lead to the modulation of signal transduction in a yet unexplored manner.

## 3. Established Drivers of the Loss of Auditory Function and Their Sensitivity to Dysregulation of the Redox System

Hearing loss can occur when any component of the auditory system is defective, and this is characterized into three distinct subtypes, depending on the locus of defects: conductive, sensorineural, and mixed hearing loss. Conductive hearing loss occurs when there is a mechanical disruption of sound transmission to the middle ear, often caused by a ruptured eardrum [77], cholesteatomas [78], otitis media [79], osteogenesis imperfecta [80], or any form of genetic mutation affecting the morphology or structure of the auditory canal up to the ossicle bones [81,82]. In situations without further auditory complications, conductive hearing loss is treated with surgical reconstruction [83,84] or managed with hearing aids [85,86]. Comparatively, sensorineural hearing loss (SNHL) happens with inner ear or auditory nerve damage and is directly amenable to environmental factors. Unlike conductive hearing loss, there is no effective treatment for SNHL once diagnosed [87]. The use of hearing aids is an unavoidable stopgap measure for compensating for existing hearing or hearing-related communication defects [88], while the control of lifestyle and environmental factors aims to delay the progression of hearing defects [87]. There is a significant proportion of patients that suffer from both physical and biological defects in the auditory system which manifest in the form of mixed hearing loss, including instances of auditory neuropathy spectrum disorder characterized by the absence of middle ear reflex and auditory brainstem response despite close-to-normal otoacoustic emission [89,90,91], originating from anatomical and genetic defects from birth or physical damage to the auditory system. From a different perspective, hearing impairment can occur as a temporary disruption of peripheral auditory function due to reversible changes to the passage of auditory signals to the inner ear or permanent loss of hearing capabilities due to the degeneration of hair cells and the auditory nerve. Interestingly, regeneration of auditory hair cells is possible only in the avian sensory epithelia [92], involving the F2R-like trypsin receptor 1 (F2RL1)/heparin-binding epidermal growth factor-like growth factor (HBEGF)/EGF receptor (EGFR)/extracellular signal-regulated kinase (ERK)-mediated re-entry of avian facultative stem cells into the S-phase of the cell cycle [93]. There is active yet nascent research on hair cell regeneration in mammals for the possibility of developing treatments to treat and cure hearing-related impairments [94,95]. Regardless of the types of hearing loss, genetic and environmental factors play an equal role in the loss of auditory function and perception.

### 3.1. Genetic Factors and Their Associated Predisposition to the Impairment of Auditory Function

The advent of DNA sequencing methods alongside classical whole-genome linkage analysis methods has allowed the systemic elucidation of the genes involved in hearing loss [96]. About half of all patients with hearing loss carry some form of genetic defect that contributes to the degeneration of auditory function [97,98], and the majority of known mutations linked to hearing impairment do not cause any other symptoms (non-syndromic) beyond auditory problems. To date, more than 200 genes are associated with hereditary hearing loss. The comprehensive and regularly updated list can be found on the Hereditary Hearing Loss Homepage (https://hereditaryhearingloss.org/, accessed on 22 February 2024). Genes linked to the development of auditory defects can be categorized based on their locus of functional impact within the auditory system, namely whether they are involved in the development and maturation of ear function or the maintenance of inner ear structural integrity for effective mechanotransduction or impact the function of the vestibular hair cells (Table 1, Figure 2).

From the preferential focus on understanding the role of identified genes in auditory function, little is known about the alternate cellular effects. An examination of the novel involvement in these genetic factors in parts of the redox pathway (Table 1, column on the speculated involvement in redox homeostasis) provides a plausible direction for future research investigating the relationship between the loss of auditory protein function, susceptibility to hearing loss, and cellular redox imbalance, particularly in the absence of cellular loss. More importantly, genetic factors with direct involvement in mitochondria function (*MTCO1* cytochrome c oxidase subunit 1 and *MTND1* reduced nicotinamide adenine dinucleotide (NADH)-ubiquinone oxidoreductase chain 1) [99,100], ROS production (*ClpP* caseinolytic mitochondrial matrix peptidase proteolytic subunit, *NLRP3* nucleotide-binding domain, leucine-rich–containing family, and pyrin domain–containing-3) [101,102], development of oxidative stress (*HSD17B4* 17-β-hydroxysteroid dehydrogenase and *WFS1* wolframin endoplasmic reticulum transmembrane glycoprotein) [103,104], and apoptosis (*AIFM1* apoptosis-inducing factor and mitochondria-associated 1) [105] link activity-dependent cellular alterations with redox imbalance and pathology of the auditory tissues.

Due to the fundamental importance of mitochondria in energy production and cellular function, mutations in the mitochondrial deoxyribonucleic acid (mtDNA) of mitrochondrial function-associated genes often involve multiple organs [106]. There is a high level of heteroplasmy in the coding regions of mtDNA in non-mitotic tissues [107], and a higher level of mutational load or the presence of modifiers may be required for development of the pathology. The expression product of *MT-CO1* (COX1) is the main subunit of cytochrome c oxidase complex I and the primary site of mitochondrial oxidative phosphorylation [108], while the associated NADH dehydrogenase function (protein encoded by *MTND1*) is essential for the oxidation of NADH, production of NAD^+^ for use in the citric acid cycle [109], and release of protons for maintenance of the mitochondrial membrane potential [110]. ClpP is a nuclear DNA-encoded peptidase and the catalytic subunit of the ClpXP complex involved in the degradation of misfolded or damaged proteins in the mitochondria [111]. By itself, ClpP serves as a mediator of the unfolded protein response [112,113] that is activated by mitochondrial dysfunction to restore electron transport chain function and remove excess ROS produced by the defective organelle [114]. While the ablation of *ClpP* in mice leads to a general upregulation of mitochondrial chaperones, accumulation of mtDNA, and increase in the expression of inflammatory markers in various organs of a *ClpP* null mouse [115], the association of ClpP with protease La (LON) is required for the removal of complex I and culling of ROS production in the presence of mitochondrial stress and when mitophagy fails [116]. In the context of energy-demanding auditory synaptic transmission, the deleterious effect of lower ATP production and ROS overproduction occurring with faulty ClpP and uncorrected complex I function [117] is likely exacerbated.

With the sustained presence of heightened ROS levels, the NLRP3 inflammasome may be activated through direct interaction with the thioredoxin-interacting protein (TXNIP) [118], driving cellular death. Within immune cells, a gain-of-function mutation in *NLRP3* is linked to the sustained release of inflammatory factors [119,120] that can impair cochlea microcirculation [121], leading to ischemic damage. Further defects in the genetic factors involved in fatty acid metabolism (*HSD17B4*) [122], endoplasmic reticulum (ER) stress signaling (*WFS1*) [123], and caspase-independent cellular death (*AIFM1*) likely contribute to the increase in ease of the activation of cell death pathways and loss of functional components of the auditry system. These associations suggest the potential for a redox imbalance to disrupt normal auditory function or the contribution of hearing-specific proteins to propagation of the cellular impact and damage with initial oxidative stress induced through alternate means.

**Table 1 antioxidants-13-00598-t001:** Genes involved in hearing loss, sorted based on the locus of impact, and their speculated links to redox imbalance and cellular function.

Gene Symbol	Protein Name	DFN Locus	Type of Hearing Loss	Normal Function	Speculated Involvement in Redox Homeostasis
Structural Defects					
*MYH14*	Myosin heavy chain 14	DFNA4	NS and S	Non-muscle ATP-dependent molecular motors interact with cytoskeletal actin to regulate cell motility and polarity [124].	miR-499 originates from the *MYH14* intronic sequence and is involved in protection of cardiomyocytes [125] and neurons [126] from tissue damage-induced oxidative stress.
*TECTA*	Alpha-tectorin	DFNA8/12/21	NS	Major non-collagenous structural component of the tectorial membrane [127].	-
*COCH*	Cochlin	DFNA9	NS	Major non-collagenous component of the extracellular matrix of the inner ear. Linked to the regulation of bacteria-driven immune response in the inner ear [128,129,130].	Shear stress which disrupts endothelial homeostasis and promotes oxidative stress leads to multimerization of cochlin and increased interaction with the mechanosensitive potassium channel subfamily K member 2 (TREK-1) in the ocular system [131].
*MYO*	Myosin, class II and III	DFNA4/11/22/48, DFNB3/30/37	NS and S	Development and function of the cochlea duct (class II) and stereocilia of the vestibular hair cells (class III) [132,133].	Myosin are differentially expressed under oxidative stress in diabetic rat brains [134].
*COL11A2*	Type XI collagen, called the pro-alpha2(XI) chain	DFNA13	NS	Minor fibrillar component of the tectorial membrane [135].	-
*CDH23*	Cadherin 23	DFNB12	NS and S	Calcium-dependent cell-cell adhesion glycoprotein involved in maintaining normal organization of stereocilia bundle [136,137].	Cadherin 23 regulates purine metabolism [138] involved in the modulation of cellular redox biology [139].
*STRC*	Stereocilin	DFNB16	NS	Structural component of the stereocilia involved in the formation of horizontal top connectors of stereocilia and maintenance of the OHC bundle [140].	-
*TRIOBP*	Trio rho guanine nucleotide exchange factor and F-actin binding protein	DFNB28	NS	Cytoskeleton-associated protein which organizes actin filaments into uniquely rootlet-like dense bundles that provide durability and rigidity to stereocilia [141].	Actin is susceptible to oxidation and effects of reactive oxygen species on its functioning [142,143,144]. Specific composition of actin may be important for stereocilia function [145,146].
*WHRN*	Whirlin	DFNB31	S	PDZ domain-containing protein expressed at the ankle region of stereocilia. Regulates IHC stereocilia growth and differentiation and OHC stereocilia rigidity and organization during development [147,148].	-
Functional Defects					
*GJB2/3/6*	Gap junction protein 2/3/6 or connexin 26/30/31	DFNA2B/3A/3B	NS	Formation of hemichannels in the sensory epithelium, required for the formation of endolymphatic potential, which create sufficient driving force for K^+^ entry and depolarization of hair cells with activation of the MET channel [149].	Connexin 26 ablation leads to increased oxidative stress in cochlea [45], likely through hemichannel-mediated spread of molecules that trigger redox imbalance in normal cells in the immediate periphery [150].
*DIAPH1*	Diaphanous homolog 1 (Drosophila) protein	DFNA1	S	Regulate actin polymerization and microtubule dynamics to stabilize the cytoskeletal structure of hair cells [151].	-
*KCNQ4*	K_v_7.4 potassium channel	DFNA2A	NS	Maintaining cochlear ion homoeostasis and regulating hair cell membrane potential [152].	-
*SLC17A8*	Solute carrier family 17 member 8 or vesicular glutamate transporter	DFNA25	NS	Involved in the uptake of glutamate into the synaptic vesicles in IHCs [153].	-
*TMC1*	Transmembrane channel-like protein 1	DFNB7/11	NS	Ion-conducting pore of the MET channel complex [154,155].	-
*SLC26A4*	Solute carrier family 26 member 4 or pendrin	DFNB4	NS	Transport negatively charged ions across the cell membrane. Involved in the function of the basal and intermediate cells of the stria vascularis to maintain the endocochlear potential [156].	Pendrin knockout (KO) in mice leads to hyperpigmentation of the stria vascularis due to the increase in pH of the endolymph, which results in inhibition of cysteine uptake and glutathione synthesis by the surrounding cells [157]. Melanin synthesis is linked to oxidative stress in melanocytes [158].
*TMPRSS3*	Transmembrane protease serine 3	DFNB8	NS	Essential component of hair cell homeostasis and key to their survival. Precise mechanism unclear [159].	-
*PJVK*	Pejvakin	DFNB59	NS	Involved in peroxisome proliferation in response to sound. Precise mechanism unclear [160].	Pejvakin-mediated pexophagy protects auditory hair cells from noise exposure-induced oxidative stress [161].
*SLC26A5*	Prestin	DFNB62	NS	Functions as the molecular motor in OHCs. Generates force of electromotility for the amplification of sound signals in OHCs [162].	Oxidative stress inhibits the expression of prestin [163].
*LHFPL5*	Lipoma high-mobility group protein gene fusion partner tetraspan subfamily member 5	DFNB67	NS	Tethers tip link to the MET channel to establish maximal force sensitivity of the MET channel. Required for correct localization of protocadherin related 15 (PCDH15) and TMC1 to the mechanotransduction complex [164,165].	-
*LOXHD1*	Lipoxygenase homology polycystin/lipoxygenase/alpha-toxin domains 1	DFNB77	NS	Involved in the mechanotranduction process in hair cells. Mechanism unknown [166,167].	-
*SERPINB6*	Serine proteinase inhibitor family B member 6	DFNB91	NS	Protect hair cells from the leakage of lysosomal content during stress [168].	Lysosomes are susceptible to oxidative stress-dependent destabilization of membrane, which leads to the release of lysosomal enzymes into the cytosol [169].
*CABP2*	Calcium binding protein 2	DFNB93	NS	Modulator of IHC Ca_v_1.3 function [170,171].	CABP2 is a thioredoxin, which contains the redox-active dithiol/disulfide bond involved in defending against oxidative stress [172,173].
Developmental Defects					
*PRPS1*	Phosphoribosyl pyrophosphate synthetase 1	DFNX1	NS/S	Catalyze first step of nucleotide synthesis. Involved in fetal auditory system development [174].	Production of nicotinamide adenine dinucleotide (NAD) is phosphoribosyl pyrophosphate (PRPP)-dependent, and pyridine nucleotides are severely reduced in erythrocytes of patients with PRPS-1 superactivity [175].
*POU3F4*	Pit-1/Oct-1/ Oct-2/unc-86 class 3 homeobox 4	DFNX2	NS	Involved in the development of the middle and inner ear [176].	The related POU3F1 is degraded in the presence of oxidative stress [177].
*EYA4*	Eyes absent transcriptional coactivator and phosphatase 4	DFNA10	S	Involved in embryonic auditory system development and mature inner ear function [178,179].	Reduced *EYA4* expression decreases single-stranded DNA accumulation following DNA damage and impairs homologous recombination [180].
*GRXCR1*	Glutaredoxin and cysteine-rich domain containing 1	DFNB25	NS	Required for the morphogenesis of stereocilia in hair cells [181].	-
*ESRRB*	Estrogen-related receptor beta	DFNB35	S	Essential for inner ear development and function [182].	ERRB is a negative regulator of NF-E2-related factor 2 (Nrf2) [183], involved in the expression of detoxifying enzyme and antioxidant proteins against oxidative stress [184,185].
*HGF*	Hepatocyte growth factor	DFNB39	NS	Involved in the development of stria vascularis of the cochlear epithelium [186].	HGF attenuates angiotensin II–induced oxidative stress in vascular smooth muscle cells [187] and protects retinal pigment epithelial cells from oxidative stress [188].
*PTPRQ*	Protein tyrosine phosphatase receptor type Q	DFNB84	NS	Essential for the maturation and function of the hair bundle in the cochlea [189].	Increase in expression of the related PTPRO increases reactive oxygen species production and promotes apoptosis through the toll-like receptor 4 (TLR4)/ nuclear factor kappa light chain-enhancer of activated B cell (NF-κB) pathway [190].

NS = non-syndromic hearing loss, not associated with alternate clinical signs or symptoms; S = syndromic hearing loss, associated with alternate clinical conditions; DFNA = genetic mutation relating to hearing loss inherited through an autosomal dominant manner; DFNB = genetic mutation relating to hearing loss inherited through an autosomal recessive manner; DFNX = X-linked heritable genetic mutation related to hearing loss.

### 3.2. Noise-Induced Hearing Loss

Chronic exposure to excessive noise can lead to mechanical stress-induced stereocilia breakage [191], destruction of the stereocilia tip links which uncouple MET channels from the movement of the stereocilia [192], or damage the F-actin core of the stereocilia, leading to a decrease in rigidity of the stereocilia [193], which contributes to the positive feedback-driven loss of mechanotransduction and sensitivity of the inner ear toward auditory signals. Although the rapid repair of tip links is possible with the temporary recruitment of shorter PCDH15/PCDH15 tip links to replace the damaged mature PCDH15/CDH23 configuration, persistent mechanical stimulation can lead to irreversible premature hair cell death [194,195]. The functional capability of the OHC to alter their length in response to changes in sound waves made them sensitive to movement-induced alteration in cellular function and cell survival [196,197]. The transient loss and reformation of the link between stereocilia and the tectorial membrane is thought to contribute in part to the temporary auditory threshold shift (TTS) observed with noise-induced auditory trauma [198,199].

It is believed that an increase in the cochlear metabolic rate after noise exposure leads to altered mitochondria metabolism and an increase in the production of reactive oxygen species (ROS) [200]. Small amounts of ROS can perform as secondary messengers to modulate intracellular signaling. For instance, physiological levels of ROS can regulate autophagy processes through protein kinase B (AKT) [201] and adenosine monophosphate-activated protein kinase (AMPK) [202] pathways, which in turn affect the clearance of clearance of oxidized cellular components and survival of hair cells [203]. In contrast, uncontrolled generation of a large amount of ROS is linked to lipid peroxidation [204], protein oxidation [205], impairment of mitochondria [206], and cell death [207]. Furthermore, altered intracellular Ca^2+^ levels may further contribute to the generation of oxidative stress. The increase in free Ca^2+^ levels observed in the cochlea following noise exposure [196] has been linked to the activity of voltage-gated calcium channels [208,209] and amounts of extracellular Ca^2+^ [210]. The excess Ca^2+^ can intersect the Krebs cycle through the calcium-sensitive α-ketoglutarate dehydrogenase to induce the generation of superoxide and hydrogen peroxide [211].

Interestingly, noise exposure alters the microcirculation and blood flow and circulation within the cochlea [212,213], likely due to the production of 8-isoprostane-F2 alpha, a lipid peroxidation product [214,215] and a potent vasoconstrictor [216]. Following the recovery of cochlear blood flow upon the termination of a noise stimulus, the inner ear experiences an upregulation of nicotinamide-adenine dinucleotide phosphate oxidase 1 (NOX1) and dual oxidase 2 (DUOX2) expression [217], increased superoxide production [218], lipid peroxidation in the hair cells and SGNs [204], and reduced levels of the antioxidant coenzymes ubiquinone-9/10 [219] and the related mitochondria dysfunction [220], which set the stage for positive feedback-driven production of ROS. The loss of dendritic spine and peripheral axons of the bipolar sensory neurons and SGNs was observed as the immediate short-term effect of noise exposure [221,222]. Strong noise exposure is further correlated with the swelling and loss of cells from the stria vascularis [223], alongside a largely transient shift in the endocochlear potential [224]. Persistent IHC stimulation contributes to the excessive release and insufficient clearance of glutamate and overstimulation of the glutamate receptors on the postsynaptic auditory nerve terminals, which allows the massive and indiscriminate influx of ions and water from the surrounding into the postsynaptic neuron, cellular swelling, dysregulation of cellular homeostasis, and eventual cellular necrosis if uncontrolled [225,226,227].

It is possible that the hair cells have a differential tolerance toward changes in oxidative stress compared with normal cell types and between the hair cells and alternate supporting cells in the cochlea [228] due to their high level of cellular activity [229], which affects the cellular response toward changes in oxidative stress. The direct exposure of mouse hair cells to high hydrogen peroxide levels activates the p53-mediated apoptosis, delay cell cycle progression in proliferating cells, and loss of nuclear structure [230], supporting the proposition that the production of ROS or changes in oxidative stress levels may lead to direct cell demise. ROS and free radicals produced with the dysregulation of redox processes can break down the cell membrane [215], leading to disruption of the cellular structure and death. The lipid peroxidation observed in cochlea cells with exposure to noise may further propagate and broaden the impact of oxidative damage on cochlea function [231]. Sensory hair cell dysregulation cumulates to activation of the caspase 3 and 9 [232], which further trigger mitochondria dysfunction, the production of ROS, and eventual apoptotic death [233].

### 3.3. Exposure to Ototoxic Chemicals and Compounds

Numerous therapeutic drugs and environmental toxicants have been shown to possess ototoxic effects through disruption of the physical structure of the peripheral auditory system or the connected neurological pathways involved in auditory perception. More often than not, the clinical or physical benefit of ototoxic drug or chemical usage is higher than their potential impact on known auditory function (if any) when the drug or chemical is first developed, which contributes to their widespread and relative lack of control of use. The aminoglycoside family of antibiotics, which comprises streptomycin, kanamycin, neomycin, and gentamicin, which is used routinely to treat gram-negative bacteria infections, rapidly enters the blood-labyrinth barrier of the stria vascularis and the cochlea upon administration and persists within the endolymph and perilymph [234,235,236]. Upon uptake into the hair cells, these antibiotics can accumulate in the lysosome, disrupt the lysosome structure, and lead to the release of lysosomal content and antibiotics within the hair cells, triggering cellular degeneration [237]. Alternatively, aminoglycosides form a complex with iron [238] which interacts with polyphosphoinositides [239] and triggers the oxidation of arachidonic acid and its byproducts present on cellular membrane structures and the production of ROS [240]. Further c-Jun N-terminal kinase (JNK)-dependent recruitment of cellular components of the apoptotic pathway can drive the cells toward death [240].

Exposure toward the platinum derivatives cisplatin and carboplatin, which are commonly used in the treatment of solid tumors, is linked to the depletion of cochlea glutathione and antioxidant activities [241] as a result of the induction of superoxide production by the NADPH oxidase (NOX) 3, a member of the NADPH oxidases which is highly expressed in the inner ear and localized to the sensory epithelia [242]. The resulting indiscriminate destruction of the inner ear’s cellular structure can turn an innocuous noise signal into a noxious stimulus that further drives the decline in auditory system function [243]. Loop diuretics (ethacrynic acid, furosemide, and bumetanide) used for the treatment of high blood pressure act on sodium- and potassium-transporters and pumps within the stria vascularis to disturb the normal ionic concentration of the endolymph, leading to disruption of the ionic gradient between the endolymph and perilymph required for maintenance of the endocochlear potential and auditory transmission [244]. Although the effects of loop diuretics often go away upon the termination of use, instances of permanent hearing loss have been observed, likely due to sustained use over long durations [245] and the drug-dependent production of free radicals with ischemia reperfusion in the lateral wall (spiral ligament and stria vascularis) [246]. High doses of acetyl salicylic acid, painkillers, and alternate antipyretics (quinine and chloroquine) have potential ototoxic side effects which are not well characterized [247,248,249,250]. Toxic interactions between the aminoglycosides, cisplatin, loop diuretics, and other drugs have been observed, which contributes to the potentiation of preexisting ototoxic effects of consumed drugs [251,252,253].

Common environmental toxicants such as aromatic solvents likely affect auditory function through the formation of reactive intermediates [254] that disrupt the membrane structure of hair cells [255] or affect K^+^ recycling in the organ of Corti. Furthermore, the organic solvents may act on specific synaptic sites [256,257,258,259] and the acoustic reflex [260,261], which suggest functional disturbance beyond the peripheral auditory system. Exposure to phytotoxic agents commonly used as bactericides and fungicides led to the development of hearing impairment and involuntary eye movements in industrial workers [262], while the administration of germanium dioxide, a component of dry-cell batteries, electrical coils, and animal food additives, in rats and guinea pigs led to stria vascularis and supporting sensory epithelia degeneration [263]. There is also evidence from animal studies pointing to the potential ototoxicity of halogenated hydrocarbons through the effect of thyroid hormones [264,265,266,267]. Further validation (human) and mechanistic studies are required to pinpoint their true effect on auditory function.

### 3.4. Impact of Inflammatory Events on Auditory System Functioning and the Age-Related Decline of Auditory Function

Otitis media (OM), or middle infection, is a common form of ear infection that can occur at any age but is most frequently diagnosed in young infants up to 24 months of age [268]. Although OM can resolve spontaneously without complications [269], chronic OM can progress toward spreading the infection into the CNS (meningitis, cerebral abscess, and encephalitis), with various degrees of permanent sensorineural hearing loss [270]. The progression from acute to chronic OM has been linked to disruption of the antioxidant system [271,272], though the implication of the effect of an altered oxidative stress or antioxidant level (if any) on the cochlea function is unknown.

Mitochondria disfunction and ROS generation are dominant drivers of nucleotide-binding domain, leucine-rich-containing family, pyrin domain-containing-3 (NLRP3) inflammasome activation [273], a critical component of innate immunity [274] in part due to the transient receptor potential melastatin 2 (TRPM2)-mediated cellular calcium influx [275]. Mitochondrial-dependent autophagy may alternatively remove the dysfunctional mitochondria, putting a stop to the aberrant ROS production [276]. However, the gain of function mutation in NLRP3 and altered inflammasome activation are involved in a spectrum of autoinflammatory conditions characterized by prominent hearing disfunction [102] linked to persistent downstream activation of the resident cochlea macrophages and the release of cytokine interleukin 1 beta (IL-1β) [119]. The suppression of macrophage recruitment [277], IL-1β release [278], or alternative inflammatory mediators such as interleukin 6 (IL6) [279] attenuates the effect of the hearing function damage by environmental insults on hearing outcomes.

With time, the accumulation of cellular damage and loss of functional sensory cells from the genetic predisposition of auditory damage or exposure to an environmental ototoxic stimulus cumulates to an unavoidable decline in hearing acuity (Figure 3). Specifically, ROS production increases, while the efficacy of the antioxidant decreases with age [280], and the associated mitochondrial disfunction and oxidative stress play key roles in the development of several age-related neurological diseases [281,282]. Mice lacking superoxide dismutase 1 (SOD1), which is critical for breaking down superoxides [283], showed premature age-related loss of sensory hair cells [284], reduced thickness of the stria vascularis, and severe degeneration of the SGNs [285], while mice with inactive glutathione peroxidase 1 (GPX1) presented a significant increase in their hearing thresholds at high sound frequencies [286]. The ROS that persist within the auditory system can damage the mitochondria DNA [287], which impairs energy metabolism [288], enhances ROS generation [289], and alters mitochondria-dependent apoptotic pathways [290]. Metabolic disorders in the form of diabetes mellites, dyslipidemia, obesity, and hypertension negatively affect the development of hearing loss with age [291] via a plethora of effects on the auditory system.

As the perception of sound is a highly neurocognitive process involving numerous brain networks directly or indirectly involved in auditory processing and representation [18,292,293], the loss of primary auditory inputs results in a major restructuring of the neurological network beyond the auditory cortices [294,295]. In addition, recent studies revealed the co-occurrence of altered auditory perception in neurodegenerative conditions [296,297,298,299], with age-related etiologies. Hence, it does not come as a surprise that there are significant associations between hearing impairment and the development of dementia and alternate neurological conditions [300], particularly neuropsychiatric conditions involving multisensory perception [301,302]. Nonetheless, their causal relationship between changes in auditory perception and alternate neurological conditions has not been clearly established.

## 4. Targeting Redox Imbalance-Driven Hearing Impairment with Antioxidants

Physiologically, the function of signaling pathways and cellular homeostasis is intimately associated with the redox balance. The endogenous antioxidant systems within involve enzymes such as vitamins A, E, and C, flavonoids such as quercetin, and trace elements such as zinc or magnesium and proteins, which exhibit antioxidant properties [303]. This has led researchers to attempt to decrease ROS levels and alleviate the effects of various antioxidants, vitamins, and more [304].

### 4.1. Natural Product-Based Antioxidant Therapies

The results focused on antioxidant therapies in ROS-related hearing loss have shown the preventive effects of antioxidants based on age, noise, and ototoxicity-induced effects [305]. ROS-related hearing loss from damage to the inner ear can be prevented by reducing the generation of ROS or by enhancing the antioxidant system, especially through the administration of exogenous antioxidants, upregulating endogenous antioxidant production, and promoting an ROS scavenger system [306]. An example of an ROS scavenger is the N-acetyl L-cysteine (NAC), which has been widely studied against noise-induced oxidative stress. This is due to its ability to directly scavenge hydrogen radicals [307].

Studies have identified various agents to protect the inner ear from oxidative stress, such as vitamin A, as mentioned previously, or even flavonoids from a clean diet of fruits or known therapeutic foods, such as ginseng and coenzyme Q10 (CoQ10), as protection against or recovering from hearing loss [305]. For this reason, the use of these exogenous antioxidant systems has been commonly tested in several animal models, with promising results. Korean red ginseng not only has anti-ROS properties but also anti-apoptotic properties. Previous studies have shown that mice fed with Korean red ginseng 1 h and 1 day after noise exposure compared with those exposed 3 days later had faster recovery from hearing loss [308].

### 4.2. Supplemental Nutrients

Regular lacking of supplemental nutrients such as magnesium, iron, and zinc was found to increase the risk of loss of hearing [309]. In auditory neurons, when zinc is located with glutamate in the presynaptic calyces and excitatory synapses as a supplement, reports have displayed recovery from hearing loss. As zinc is an essential metal ion in physiological processes, especially as an antioxidant in Zn^2+^ form, it maintains the structural integrity and functions of DNA and proteins [310]. Regarding its antioxidant role, it inhibits the oxidization of NADPH, preventing the generation of ROS. Zinc also plays an important role as a cofactor of copper/zinc superoxide dismutase-1 (Cu/Zn SOD; SOD1), called the cell’s first line of antioxidant enzymes. It was reported to exist while bounded with SOD1 in rat cochlea, illustrating the elevation of the auditory brain stem (ABR) threshold and cochlear hair cell loss [311].

### 4.3. Ototoxic and Novel Drugs

Based on research, two types of ototoxic drug classes are used in clinical practice, which are aminoglycoside antibiotics and platinum-based anticancer drugs [312]. These have displayed damage to the hair cells in the organ of Corti through ROS production in the apoptopic pathways. Close monitoring of their potential ototoxicity and nephrotoxicity have been previously carried out [313]. Although the cells of the proximal convoluted tubules of the kidney can proliferate and recover from nephrotoxicity, the hair cells in the cochlea are not able to recuperate and recover from irreversible ototoxicity.

In addition to that, platinum-based anticancer drugs that include adenocarcinoma, squamous cell carcinoma, and undifferentiated carcinoma have toxic effects on the cochlea and neurons [314]. Cisplatin-related ototoxicity was reported to activate ROS production in the inner ear [315]. It upregulates NOX3 expression, activates its signaling pathway, and increases superoxide production in cultured cells and the cochlea [316]. Thus, disruption of the intracellular antioxidant system also contributes to both aminoglycoside- and cisplatin-induced oxidative stress [317].

However, there is still a lack of information on the fundamental mechanisms of redox homeostasis and molecular redox networks. In light of these pushbacks, there is only one clinically available drug made in 2022 that was approved by the Food and Drug Administration (FDA) of the United States, which is sodium thiosulfate. This drug acts as an antioxidant and a therapeutic agent, based on clinical trials [318,319]. Clinical tests showed successful rates of reduced cisplatin-induced ototoxicity of nearly 50% in hepatoblastoma patients. It also preserved the activity of antioxidant enzymes and alleviates hearing loss after 4–6 h of administration without interfering with the antineoplastic activity of cisplatin [320]. Nevertheless, the therapeutic effect was limited to patients under 18 years of age, while effects varied depending on the time intervals between the administration of cisplatin and sodium thiosulfate [305]. Therefore, the development of a more effective treatment for a wide range of acquired hearing loss is still crucial.

## 5. Conclusions

The auditory outcome depends on a myriad of factors that control the function of individual nodes for auditory signal transduction and interact with each other to maintain optimal primary auditory system functioning. Interestingly, regardless of the locus of change, the disruption of the redox system or the generation of oxidative stress serves as the mechanism for the convergence of system and cellular disturbances toward hair cell and sensory epithelial loss and overall auditory function disruption. Numerous reviews have attempted to summarize existing strategies to combat problems with the dysregulation of cellular redox homeostasis in hearing loss [44,304,321,322], which falls short of an understanding of the optimal time and duration for treatment, since overtreatment with antioxidants is potentially more problematic than beneficial [323]. It is also unclear and a general problem in the field of genomic sequencing and disease mapping how pathogenic variants of a single gene may contribute similarly or differently to a disease phenotype. Much more work is required to resolve the contribution of pathogenic mutations to environmental factors in the development of hearing loss and understand the different impact on auditory function and the relationship with the time of phenotypic appearance, how alternate non-auditory disease may delay or exacerbate the development of hearing loss, and finally how the development of an understanding of the different involvement of oxidative stress in the disruption of different auditory cell types (sensory or supporting) may educate us regarding cell-dependent vulnerability to insults and demise.

## Figures and Tables

**Figure 1 antioxidants-13-00598-f001:**
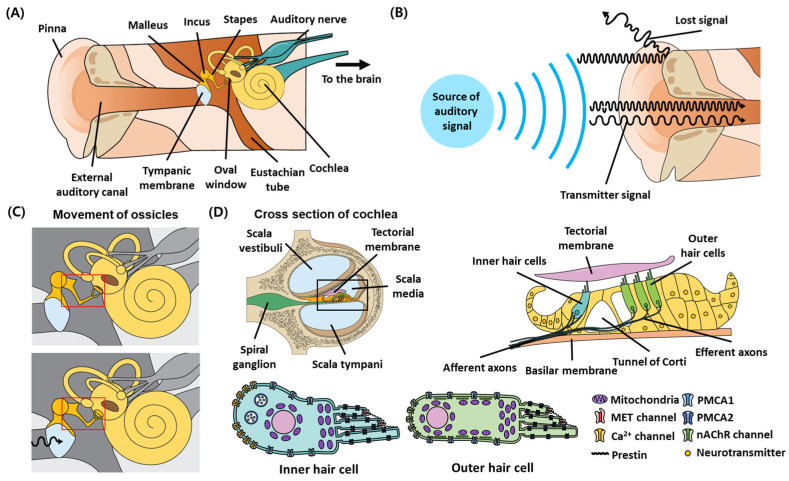
The anatomical and biological structure of the auditory system. (**A**) The anatomy of the auditory system comprises the outer auditory canal (outer ear), auditory ossicles (middle ear), and the cochlea (inner ear). (**B**) The outer ear canal forms a physical passageway which collects and channels external sound signals into the middle ear. (**C**) The sound waves are made up of high-amplitude vibrations of air particles (represented by a black waveform with an arrow showing the direction of movement) which displace the eardrum from its basal position (top compared with bottom figure, red box). The physical connection between the eardrum, ossicles, and tympanic membrane allows physical conduction of the movement of the eardrum to the tympanic membrane and the inner ear of the auditory system. (**D**) The inner ear is made up of the fluid-filled, spiral-shaped cochlea, which contains the mechanosensory hair cells required for the transduction of auditory signals into the CNS (top left). The stereocilia on hair cells are connected to the tectorial membrane through the calcium-rich filamentous structures that allow the transmission of fluid-dependent movement of the tectorial membrane to the mechanosensitive mechanoelectrical transduction (MET) channel of the hair cells. Activation of the sensory hair cells leads to the release of neurotransmitters targeting the innervating auditory nerve for transmission of auditory signals into the CNS (top right). The inner and outer hair cells comprise slightly different molecular components, which allow their specialized functions (bottom).

**Figure 2 antioxidants-13-00598-f002:**
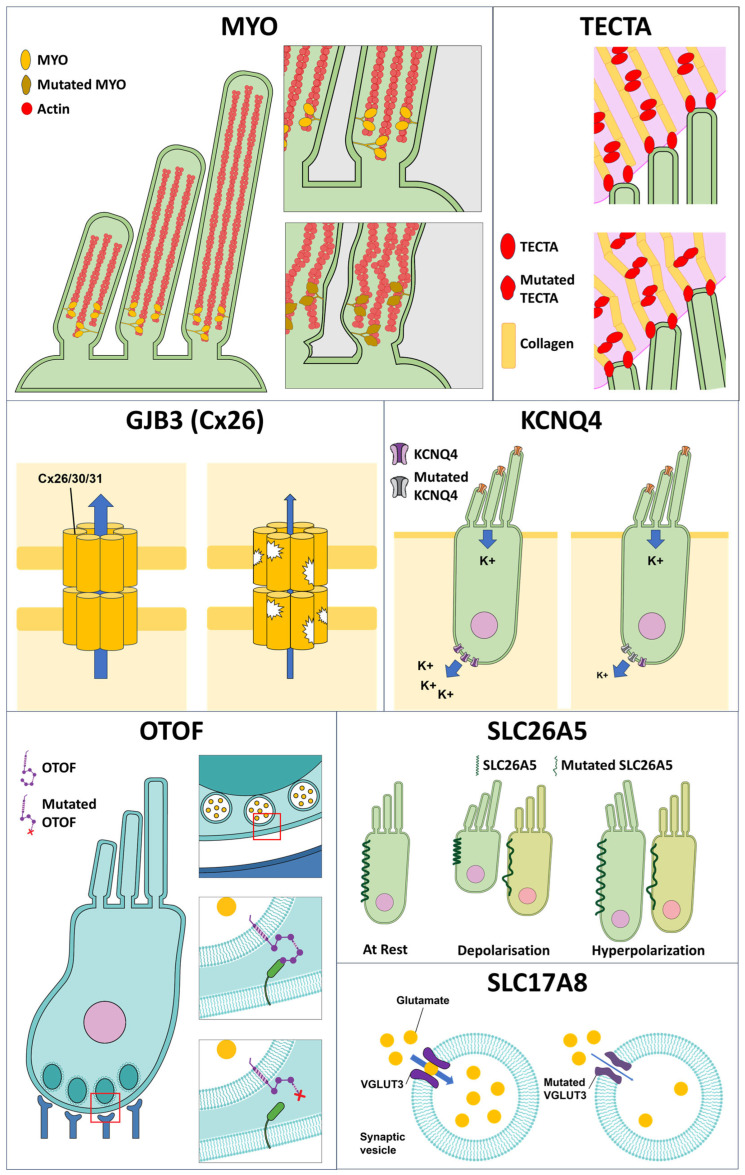
The structural and functional impact of hereditary mutations in genes involved in auditory function. Myosin VI (Myo6) and alpha-tectorin (TECTA) are involved in stereocilia structure maintenance. Mutations in Myo6 and TECTA lead to the mislocalization of the actin (MYO, bottom right) and the loss in alignment of the stereocilia structure to the tectorial membrane (TECTA, bottom right). Alternatively, mutations in the functional components of the hair cells can lead to significant alteration of the ionic gradient required for maintenance of the endolymphatic potential (GJB, right) and hair cell membrane potential (KCNQ4, right), which impacts the ease of hair cell activation and signal transmission to the auditory nerve. OTOF and SLC17A8 mutations are linked to a reduction in the quantal size of the neurotransmitter released (SLC17A8, right) and loss of the abilities of vesicle fusion and neurotransmitter release (OTOF, bottom right) upon IHC activation. Changes to the OHC electromobility (SLC26A5, depolarization and hyperpolarization) further reduce the sensitivity of the sensory hair cells toward stimulus-evoked activation. The contributions of individual effects or the combinatorial effect of the altered structural and functional components reduce the sensitivity and, in severe cases, ablate stimulus-dependent hair cell activation, leading to hearing impairment.

**Figure 3 antioxidants-13-00598-f003:**
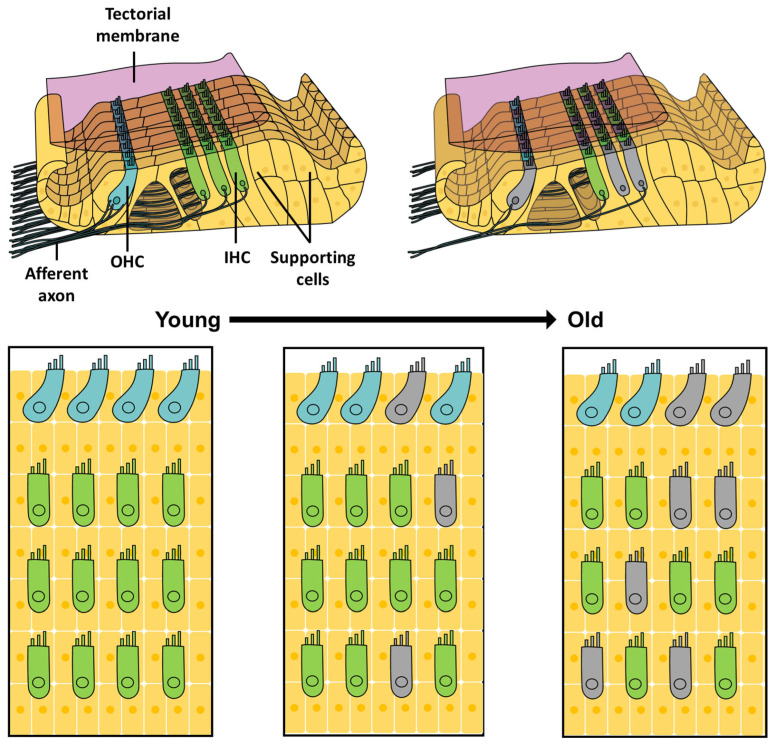
Progressive loss of cells and auditory function with time. Genetic predisposition to auditory disfunction and cumulative exposure toward ototoxic stimulus leads to the gradual loss of sensory hair cells over time.

## Data Availability

Not applicable.

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
