# Peer review of "A Consolidated Understanding of the Contribution of Redox Dysregulation in the Development of Hearing Impairment"

_antioxidants, 2024, doi:10.3390/antiox13050598_

Round 1

Reviewer 1 Report

In this interesting review article, the authors discuss the role of redox imbalance in hearing disorders, e.g., noise-induced hearing loss, or drug-induced hearing loss. Although quite simplistic in its approach, the review is quite comprehensive, allowing readers outside the field to engage with it.

Table 1: The purpose of this table is not quite clear. The citations provided to support redox imbalance are sometimes quite tenuous, especially for hearing disorders, e.g., the citations that are provided to support TRIOBP, DIAPH1. Please provide a discussion of how the (presumed mutant forms of) the proteins are linked to oxidative stress in hearing conditions. For example, is there a specific expression in hearing-related cells? Are there other phenotypes (other than hearing deficits) associated with these mutant protein forms? Or, are the phenotypes specifically with regard to hearing?

What about other genes that when mutant, cause hearing loss and enhance susceptibility to hearing loss, e.g., WFS1?

The review deserves a section on anti-oxidants and their effect on hearing disorders, e.g., NSAIDs.

Smaller points

Line 52 - the meaning of the phrase "(air, bone, or soft tissue)" is not quite clear.

Reviewer 2 Report

I appreciate the opportunity to review the manuscript for publication in MDPI Antioxidants.

I feel that the topics are interesting, and the manuscript is grossly well organized.

I have a few comments as follows.

L130: In Figure 1C, “The sound waves are made up of high-amplitude vibrations of air particles which displace the eardrum from its basal position (represented by Top schematic). The physical connection between the eardrum, ossicles, and the tympanic membrane allows the physical conduction of the movement of the eardrum to the tympanic membrane (Bottom) and the inner ear of the auditory system.” It is rather difficult to discriminate the difference between the 2 illustrations. Pointing with arrows would be helpful.

In Table 1, the authors had better add footnotes to explain abbreviated terms more in detail.

The search strategy, study screening and selection, and data extraction were performed systematically and scientifically. The data reporting and discussion were also adequately done.

See above.

Reviewer 3 Report

Overall, the main weakness is lack of focus and depth in linking genetics and environmental factors to redox imbalance. The introduction needs expansion and specific examples need more details. Addressing these will significantly improve the quality of the review.

I have the following suggestions to revise this scientific manuscript:

Abstract:

- The abstract would benefit from briefly stating the main focus on redox imbalance as a converging factor in hearing loss.  

- Avoid vague terms like "complete understanding is a work-in-progress" - be more specific about gaps.

- Briefly mention 1-2 key examples linking genes/environmental factors to redox dysfunction.

Introduction:

- Expand more on the background and prior evidence implicating redox imbalance in hearing loss to establish rationale. cite

- State main objective/focus of review clearly - as is, it's vague.

Genetics section:

- Provide more details for 1-2 gene examples - how mutations cause hearing loss, evidence linking to redox imbalance. Avoid vague statements.

- Table 1 seems speculative - need to focus on established evidence. Would move general redox links to introduction.

Noise-induced hearing loss:

- Expand on specific pathways of oxidative damage - ROS production, lipid peroxidation etc. Cite key studies.

- Provide quantitative evidence of redox changes with noise exposure from prior literature. Avoid vague terms like "massive influx".

Round 2

Reviewer 1 Report

I thank the authors for attending to all of my comments.

All of my comments and requested clarifications have been well addressed.

Author Response

We thank the reviewer for the time and effort invested towards improving our manuscript. 

Reviewer 2 Report

I reckon that the manuscript has been revised and improved in accordance with the reviewers’ comments.

None

Author Response

We thank the reviewer for the time spent reviewing our manuscript.

Reviewer 3 Report

Abstract

Rewrite the abstract to make sure it expresses the goal, methodology, important discoveries, and conclusions of the review in plain terms. Make sure to communicate your important points succinctly.

Introduction:

To create a more concentrated narrative that leads to the purpose and goals of this review, narrow the scope of the background.

- To reinforce the background, include one or two extra references on pertinent ideas such as redox biology. cite https://doi.org/10.3390/life14040425

- Include a paragraph at the conclusion that explains the review's goals, purpose, and hypotheses in detail.

Methods

Include a brief section on methodology that details the literature search strategy, including databases, search terms, inclusion/exclusion criteria, and the screening and selection procedure for studies.

Results

- Reorganize the results section such that each main idea or finding has its subsection.

- Recap the main findings rather than going into depth about each study. To draw attention to the key findings, use tables and figures.

- Eliminate any material that is repeated in the background and results sections. Discussion: Go beyond simple summarization and address implications, constraints, and potential future directions based on the evaluated studies. Provide a deeper interpretive synthesis of the data.

- To enhance the flow between issues, use subheadings.

- Include a paragraph at the conclusion that summarizes the key lessons learned and the importance of the review.

Discussion:

- Go beyond simple summarization and address implications, constraints, and potential future directions based on the evaluated studies. Provide a deeper interpretive synthesis of the data.
- To enhance the flow between issues, use subheadings. cite
doi:10.3389/fcell.2023.1119773.
- Include a paragraph at the conclusion that summarizes the key lessons learned and the importance of the review.

- discuss all the potential mechanisms of autosomic genetic hearing loss.  cite doi:10.3390/biomedicines11061616.

no
